

# Hydrodynamic and Atmospheric Conditions in a Volcanic Caldera: A Comprehensive Dataset at Deception Island, Antarctica

Francesco Ferrari[1, 2], Carmen Zarzuelo[3], Alejandro López-Ruiz[3], and Andrea Lira-Loarca[1]

[1]Department of Civil, Chemical and Environmental Engineering. University of Genoa, Via Montallegro 1, 16145, Genoa, Italy
[2]Istituto Nazionale di Fisica Nucleare, Sezione di Genova, Via Dodecaneso 33, 16146, Genoa, Italy
[3]Departamento de Ingeniería Aeroespacial y Mecánica de Fluidos, Universidad de Sevilla, Camino de los Descubrimientos s/n, 41092, Seville, Spain

**Correspondence:** Carmen Zarzuelo (czarzuelo@us.es)

**Abstract.** Marine spatial planning and environmental management in Antarctica require reliable data to address challenges such as climate change impacts, sea level changes and the dynamics of fragile ecosystems. Deception Island, a volcanic caldera in the South Shetland Islands, presents unique hydrodynamic conditions influenced by extreme weather, glacial melt and its complex geomorphology. To improve understanding of these processes, we present an open-access, integrated dataset spanning

16 years, from 2005 until 2020, combining high-resolution atmospheric and hydrodynamic variables. Atmospheric modeling was done with the Weather Research and Forecasting model leading to data in a 1-km grid and 35 vertical levels covering the island. The atmospheric dataset includes a total of 161 variables including wind fields, precipitation, pressure, among others at hourly resolution that have been validated against data provided by an in-situ weather station.

Hydrodynamical and wave propagation modeling was performed with DELFT3D on different grids with a maximum reso-

10 lution of $15 \times 25$ m$^2$ for hydrodynamic and $220 \times 160$ m$^2$ for wave propagation results. This dataset provides high-resolution temporal and spatial data including sea surface elevation, current velocities, significant wave height, wave direction and wind pressure, at daily intervals across the grid and hourly at five observation points. In addition to standard conditions, the dataset captures spatial, seasonal and temporal variability as well as extreme events, providing unprecedented insight into the island's dynamics.

By incorporating long-term high-resolution atmospheric reanalysis and hydrodynamic simulations, this dataset fills critical knowledge gaps about the hydrodynamic behaviour of Deception Island and provides a valuable tool for stakeholders in research, environmental monitoring and climate change adaptation. Applications range from analysing glacial melt contributions and nutrient transport to modelling ecosystem interactions and assessing the impacts of extreme weather events. This comprehensive data collection advances our understanding of Antarctic coastal systems and supports broader efforts to predict and

mitigate the effects of global climate change on polar environments.

## 1 Introduction

Deception Island is a unique Antarctic environment characterised by complex hydrodynamics, volcanic activity and extreme meteorological conditions. Despite its importance, detailed long-term hydrodynamic studies remain limited, particularly re-



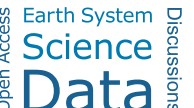

garding the role of extreme events in shaping its coastal and sedimentary dynamics. This study presents a high-resolution
dataset integrating atmospheric simulations using the Weather Research and Forecasting (WRF) model with hydrodynamic
simulations using Delft3D. By providing long-term spatial, seasonal and extreme event analyses, this work allows a better
understanding of the island's response to ocean-atmosphere interactions.

The paucity of detailed hydrodynamic data underscores critical knowledge gaps in the understanding of Deception Island's
environmental processes, particularly their interactions with atmospheric, geological and ecological factors. The integration
of the Weather Research and Forecasting (WRF) atmospheric model and data, with their unprecedented temporal and spatial
resolution, provides new insights into the long-term trends and extreme events that shape the hydrodynamics of this region.
This work builds on and complements previous studies, such as Baldwin and Smith Jr (2003); Flexas et al. (2017), which
analysed sediment transport and coastal dynamics, and Geyer et al. (2021), which investigated the contribution of glacial
melt to local hydrodynamics. In addition, Jigena et al. (2015) highlighted the role of guano-derived nutrient enrichment in
Antarctic ecosystems, and Torrecillas et al. (2024) provided broader insights into polar hydrodynamics, which can now be
further contextualised using the presented dataset.

The use of physically based numerical modelling, supported by wind speed and pressure fields from the WRF atmospheric
model, and, wave boundary conditions from the E.U. Copernicus Marine Service, provides a robust approach to studying
the hydrodynamics of the island. The dataset allows detailed analyses of wind, wave and atmospheric pressure variability
on hydrodynamic processes and provides a basis for scenario testing under extreme events or climate-induced changes. In
addition, this work facilitates interdisciplinary research by linking hydrodynamics with sediment transport, nutrient cycling
and ecosystem dynamics.

By addressing significant knowledge gaps, this dataset provides an innovative resource for advancing understanding of
the hydrodynamics of Deception Island and contributes to broader studies of Antarctic coastal and marine systems. Future
applications include assessing the effects of glacial melt on sediment and nutrient transport, analysing water quality dynamics,
and evaluating ecosystem responses to environmental and climate change. This work demonstrates the usefulness of integrating
long-term atmospheric reconstructions and high-resolution hydrodynamic modelling to improve the predictive power of models
and promote a deeper understanding of the interconnected processes that shape the Deception Island environment.

## 2   Area description

Deception Island is a volcanic caldera in the South Shetland Islands off the Antarctic Peninsula that forms one of the world's
most unique natural harbours (Figure 1). The horseshoe shaped caldera of the island, which is approximately 12km in diameter,
contains a bay known as Port Foster. This harbour, with an average depth of 100-180 m, serves as a well-protected and ice-free
harbour, ideal for maritime operations in the otherwise challenging Antarctic waters (Orheim, 1982; Smellie et al., 2002). The
island is characterised by a cold polar climate, with temperatures ranging from -10°C to 3°C, and a landscape dominated by
volcanic ash, lava flows and glacial remnants (Birkenmajer, 1992).




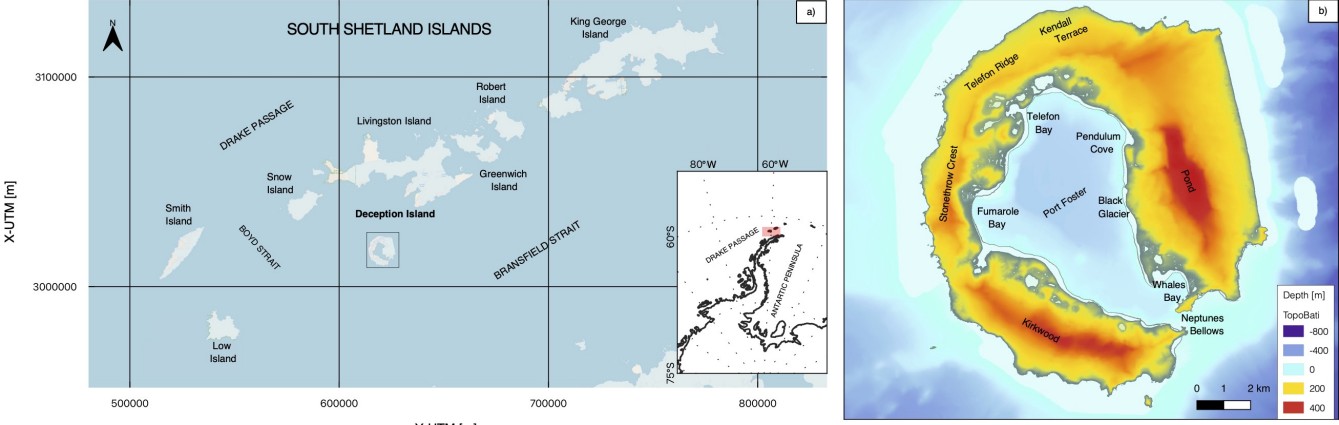

**Figure 1.** Location and topo-bathymetry of Deception Island, Antarctica

The volcanic history of the island has resulted in an active geothermal environment, particularly along the south-western coastline near Whalers Bay. Here the shallow waters are often warmed by volcanic heat, creating fumaroles, warm beaches and occasionally steam rising from the shores. The caldera has erupted several times in recorded history, with notable eruptions in 1967, 1969 and 1970, which significantly altered the island's topography and affected the scientific bases stationed there (Baker et al., 1975; Smellie, 2001). As a results, Deception Island's unique geological and environmental conditions support a limited but specialised ecosystem. The coastal areas support diverse seabird colonies, including Antarctic terns, skuas and chinstrap penguins, while the surrounding waters support seals and whales (Angulo-Preckler et al., 2021). The island's geothermal activity also supports the growth of certain thermophilic microorganisms in areas of active fumaroles (Herbold et al., 2014).

Human presence on Deception Island has largely been driven by scientific research and tourism. The island was once home to a whaling station at Whalers Bay, which has since been abandoned, but remnants of the station remain as historical artefacts. In recent decades, Deception Island has become an important site for Antarctic research, with its caldera serving as a natural laboratory for studying volcanology, oceanographic dynamics and the effects of climate change on Antarctic ecosystems (Smellie, 2001; Convey and Peck, 2019).

Port Foster can be divided into distinct zones based on bathymetry and exposure to volcanic activity. The outer caldera rim, with cliffs rising to 540 m, encloses the bay and protects it from the harsh winds and waves of the Southern Ocean. The inner bay, approximately 9 km in diameter, is connected to the open ocean by a narrow passage called Neptune's Bellows, which is 230 m wide and only 15 m deep (Smellie et al., 2002; Flexas et al., 2017). This narrow passage amplifies tidal currents, which can reach up to 1.54 m·s$^{-1}$, creating strong currents as water is exchanged between the caldera and the surrounding ocean (Flexas et al., 2017). The wind regime at Deception Island is dominated by strong westerly winds, which can exceed 20 m·s$^{-1}$ during storms. These winds generate waves that typically range from 1 to 2 m within Port Foster, but waves from the Southern Ocean entering through Neptune's Bellows can reach heights of 3 m (Figueiredo et al., 2018). These wave patterns, together with tidal influences, make the entrance challenging for ships, especially in rough weather conditions. Water levels

in Port Foster are influenced by both tidal and atmospheric pressure changes. The tidal range is typically between 1 and 2 m, but can increase during storm surges, particularly when low pressure systems pass through the area (Flexas et al., 2017). The combination of wind-driven waves, tidal surges and currents results in dynamic water conditions, particularly along the eastern and northern coasts of the island.

## 3   Material and Methods

This study presents a comprehensive data of atmospheric and hydrodynamic conditions produced using the WRF model and Delft3D, respectively (Ferrari and Lira-Loarca, 2025; Zarzuelo et al., 2025a). The WRF atmospheric model was forced using ERA-5 reanalysis whereas Delft3D was forced using the wind and pressure fields from WRF dynamical downscaling and wave data from the Copernicus Marine Service. This section presents the details on the forcing data used throughout the modelling chain and the description, set-up, calibration and validation of the atmospheric and hydrodynamics simulations.

### 3.1   Forcing Data

#### 3.1.1   ERA5

Initial and boundary conditions needed to create the atmospheric high resolution hindcast dataset were taken from the European Centre for Medium-Range Weather Forecasts (ECMWF) Reanalysis, version 5 (ERA5). ERA5 is the fifth-generation, state-of-the-art, atmospheric reanalysis of the global climate, covering the period from January 1950 to the present. It is produced by the Copernicus Climate Change Service (C3S) at the ECMWF. ERA5 provides hourly estimates for a wide range of atmospheric, land, and oceanic climate variables. The dataset covers the entire Earth on a 30 km resolution grid and models the atmosphere with 137 vertical levels, extending from the surface up to a height of 80 km (Hersbach et al., 2020). The ERA5 new reanalysis replaces the ERA-Interim reanalysis, which spanned from 1979 onwards and began in 2006. ERA5 is based on the Integrated Forecasting System (IFS) Cy41r2, that is operational since 2016, resulting hence in a decade of advancements in model physics, core dynamics, and data assimilation. Furthermore, ERA5 spatial resolution has been significantly improved with respect to 80 km horizontal resolution of the ERA-Interim products (Hersbach et al., 2020).

#### 3.1.2   Global Ocean Waves Reanalysis

The wave forcing for the hydrodynamic model was obtained from the Global Ocean Waves Reanalysis provided by the E.U. Copernicus Marine Service Information (CMEMS) -Product ID: GLOBAL_REANALYSIS_WAV_001_032. This dataset offers a high-resolution reanalysis of global ocean wave conditions since 1980, assimilating altimeter wave observations into the wave model from 2017 onwards. This dataset provides wave parameters such as significant wave height, mean and peak wave period, and mean wave direction with 0.2° spatial resolution and 3-hour temporal resolution. The dataset ensures consistency with historical observations and is widely used for ocean modeling applications, coastal engineering studies, and climate variability assessments.



## 3.2 Atmospheric modelling

### 3.2.1 Model description and set-up

To accurately describe the weather evolution on Deception Island, and specifically to precisely depict the wind and pressure fields within the island bay, which is necessary to initialize the hydrodynamic models used in this study, the ERA5 data are not optimal. The island, in fact, has an extension of about 10 km both in the north-south and east-west directions, which is too small for its interaction with the atmosphere to be accurately represented in the ERA5 reanalysis, as these have a resolution of approximately 30 km. A dynamic downscaling of the ERA5 reanalysis was therefore performed to produce a

high-resolution hindcast of the area under study, accounting for small-scale atmospheric interactions both with Deception Island and the surrounding lands. To dynamically downscale ERA5 reanalysis the Weather Research and Forecasting (WRF) model, Version 4.3.3, was adopted. The WRF model is a fully compressible non-hydrostatic, primitive-equation model with multiple nesting capabilities. WRF model represents the state of the art in numerical modeling of the atmosphere and a comprehensive description of the model formulation is given in Skamarock, 2008.

To describe in detail the atmospheric evolution around Deception Island, two nested computational domains were defined, respectively covering northernmost part of the Antarctic Peninsula with horizontal resolution of 5.0 km, and South Shetland Islands with a grid spacing of 1.0 km. Exact extension, localization and topography of the two domains are reported in Figure 2. The number of terrain-following vertical levels adopted was 35 for both domains, with higher resolution close to the surface. In Figure 2, the highlighted-inlet does not represent a computational domain, but the area where the results will be made

publicly available.

To describe the effects that unresolved sub-grid phenomena have on resolved variables, several parameterizations are available in WRF. For the present work, we adopted a model configuration quite similar to that described in Xue et al. (2022), where an assessment of WRF simulations reliability in Antartic region is performed. In particular, both for the long and short-wave radiation the Rapid Radiation Transfer Model (RRTM) scheme was selected (Clough et al., 2005). The Mel-

130 lor–Yamada–Nakanishi–Niino (MYNN) 2.5-level scheme (Nakanishi and Niino, 2006) has been selected to parametrize the Planetary Boundary Layer (PBL) as well as the MYNN scheme for the Surface Layer (SL) (Olson et al., 2021). The unified Noah land surface model was instead adopted to describe land-surface processes. Regarding the cloud microphysics parameterization, the Thompson microphysics (Thompson et al., 2004), a well-known and widely tested two-moment bulk scheme, considering all six hydrometeors, was chosen (Cassola et al., 2015). Finally, convection was explicitly resolved over the higher

resolution domains, while the Grell-Freitas cumulus parameterizatoin scheme (Grell and Freitas, 2014) was adopted for the 5 km resolution domain.

The WRF simulations cover the period from January 1st, 2005 until December 31st 2022 providing 24-h-long runs that were initialized at 00 UTC of each day from ERA5 data, while boundary conditions where imposed every 3 hours.

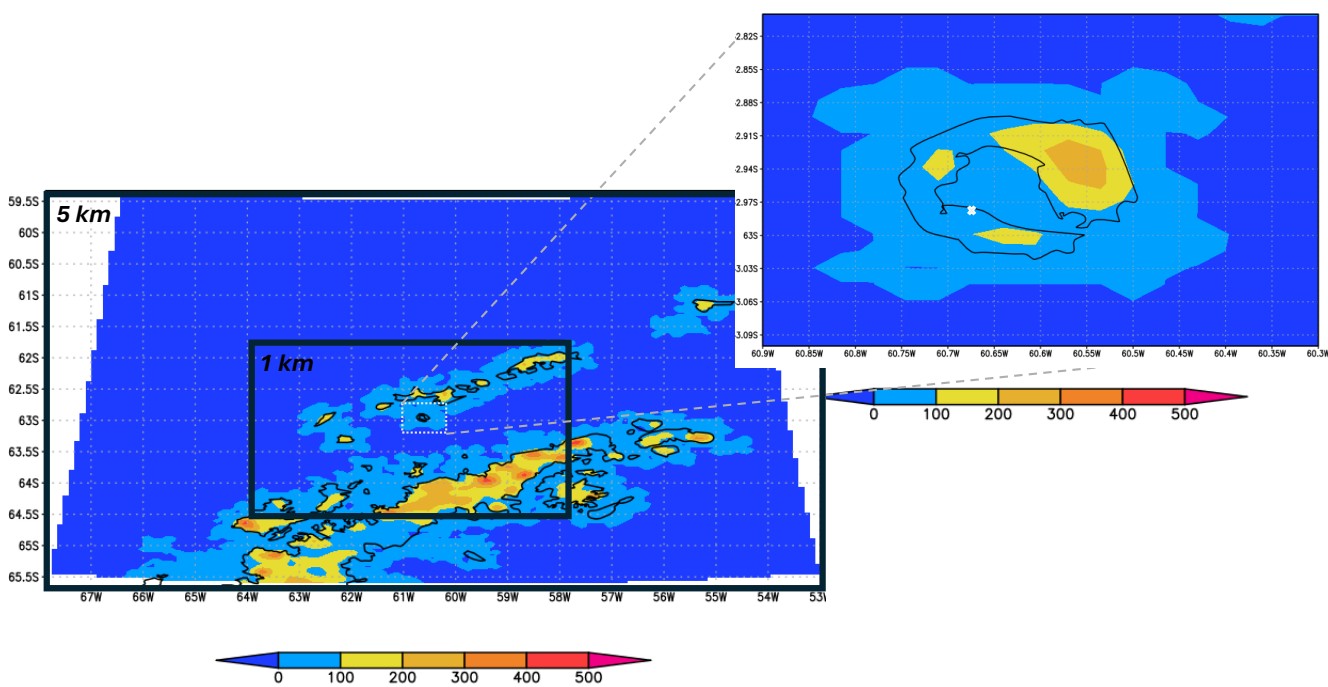

**Figure 2.** Extension, localization and topography of the two WRF nested domains. Resolution varies from 5 km for the outermost domain, to 1 km for the innermost. The highlighted-inlet does not represent another computational domain, but the area where the results will be made publicly available. The white cross in the inlet represents the location of the Gabriel de Castilla (GdC) meteorological station.

### 3.2.2 Model validation

The validation of the atmospheric downscaling modelling was carried out by comparing its outputs with observational data from the Spanish State Meteorological Agency (AEMET) meteorological station at the Gabriel de Castilla (GdC) station located in Deception Island (60°40'31" W, 62°58'38" S, white cross in the inlet of Figure 2) at an elevation of 13 m above sea level and active since February 2005. This location provided a reliable source of in-situ measurements of weather variables, allowing for a robust evaluation of the performance of WRF data. More specifically, we compared wind speed (WRF: 10-m

wind speed), wind gusts, temperature (WRF: 2-m temperature), pressure, relative humidity and precipitation between the GdC station and the closest WRF model point at 4.5 m above MSL from 05/02/2005 until 31/12/2022. Figure 3 presents the bivariate density function as well as the correlation coefficient ($R$), the Bias and the Root-mean-square-deviation ($RMSD$) for all the analyzed variables. For the precipitation, the comparison was done for the data limited to temperatures over 1 degree and daily-accumulated precipitation over 1 mm. The comparison revealed that the WRF model captured the general trends observed in the

data, with correlation coefficients ranging from $R = 0.43$ and $R = 0.57$ for the precipitation and relative humidity, respectively, to $R = 0.86$ and $R = 0.99$ for the temperature and pressure, respectively. The comparison demonstrated the accuracy of the

Earth System
Science
Data

dataset in representing real-world conditions in Deception Island. The lower correlation values for precipitation could be due inaccurate measurement of the rain gauge in case of solid precipitation, related to the mechanic characteristics of the rain gauge, and/or to the dynamic interactions between snow, winds and gauge that lead to possible differences, in certain cases, between observations and model.

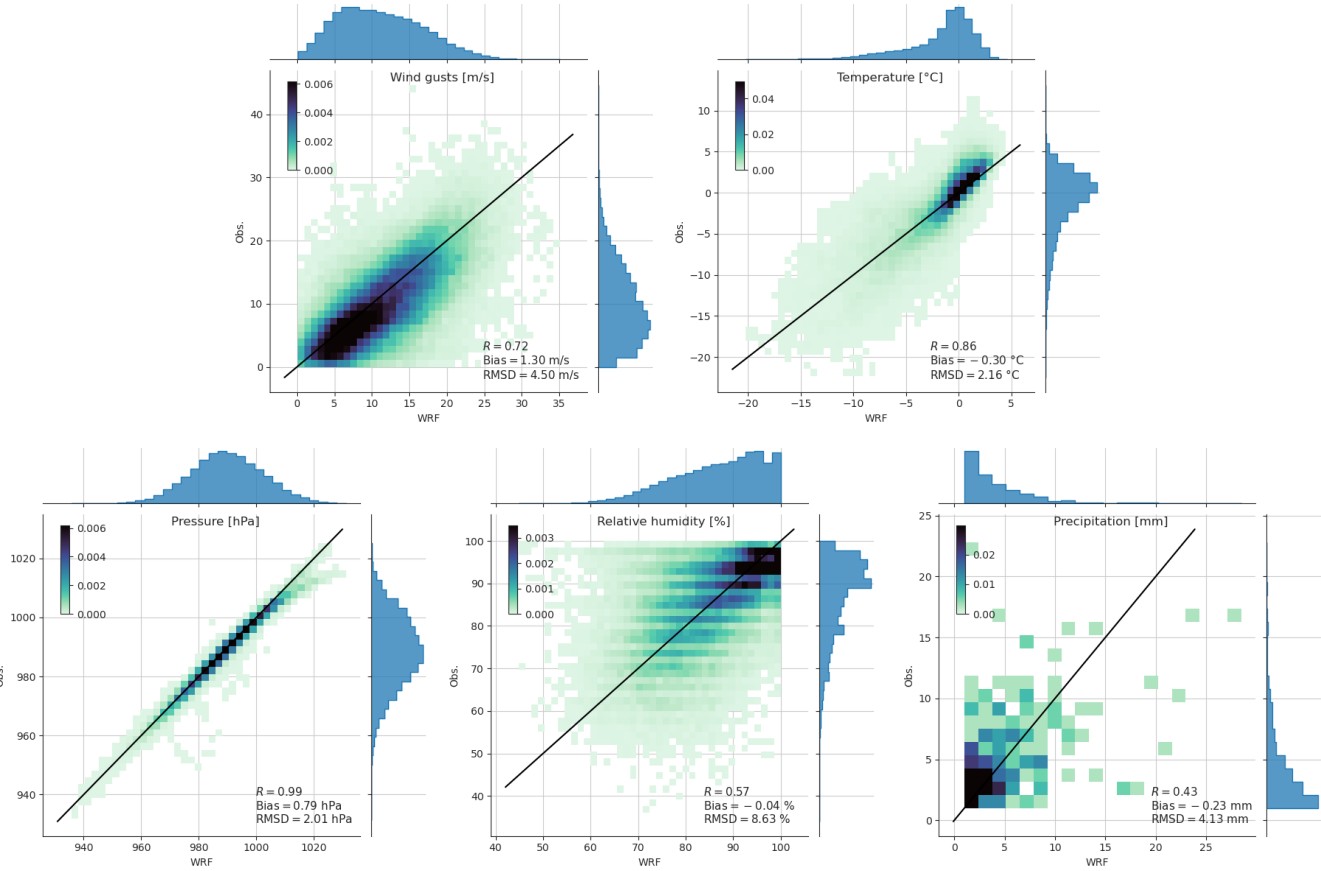

**Figure 3.** Comparison between the WRF model outputs and the observations from the AEMET GdC meteo station for wind speed, wind gusts, temperature, pressure, relative humidity and precipitation.

## 3.3 Hydrodynamic modelling

### 3.3.1 Model description

Delft3D is a widely used numerical modeling system for simulating hydrodynamic, sediment transport, and morphodynamic processes in coastal, estuarine, and riverine environments. Developed by Deltares, the model consists of several interlinked



modules, each capable of handling specific physical processes. Among them, modules Flow and Wave were used in this work
to obtain wind, pressure, tidal and wave generated hydrodynamics.

The Delft3D-Flow module is designed to simulate hydrodynamic processes, including currents, water levels, and transport phenomena driven by tidal forces, winds, and density gradients. It solves the unsteady, depth-averaged or fully three-dimensional forms of the Navier-Stokes equations, under the shallow water assumption, for free surface flows. The governing

equations can be expressed as continuity and momentum equations:

$$\frac{\partial h}{\partial t} + \nabla \cdot (h\mathbf{u}) = 0 \tag{1}$$

where h is the water depth and $\mathbf{u}$ is the horizontal velocity vector (u,v).

$$\frac{\partial (hu)}{\partial t} + \nabla \cdot (hu\mathbf{u}) = -h\frac{\partial p}{\partial x} + hfv - \tau_x + R_x + M_x \tag{2}$$

$$\frac{\partial (hv)}{\partial t} + \nabla \cdot (hv\mathbf{u}) = -h\frac{\partial p}{\partial y} + hfu - \tau_y + R_y + M_y \tag{3}$$

where u and v are the depth-averaged velocity components in the x- and y-directions, respectively; p is the pressure; f is
the Coriolis parameter; $\tau_x$ and $\tau_y$ are the wind stress components in the x-and y-directions, respectively; $R_x$ and $R_y$ are the
diffusion and dispersion terms in the x-and y-directions; and and $M_x$ and $M_y$ represent contributions due to external sources
or sinks of momentum, such as wind action on the water surface or wave action in the water column. The model incorporates
various turbulence closure schemes such as the k-$\epsilon$ model, which helps in capturing vertical mixing and turbulent energy

dissipation (Rodi, 2017). Delft3D-Flow is commonly applied in coastal engineering studies to analyze storm surges, tidal
dynamics, riverine flows, and long-term morphological evolution (Lesser et al., 2004; Zarzuelo et al., 2021).

The Delft3D-WAVE module is based on the SWAN (Simulating WAves Nearshore) model, which computes the transformation of surface waves as they travel across the coastal region. It simulates processes such as wave generation by wind,
wave propagation, refraction generated by depth variations and currents, diffraction, and nonlinear wave-wave interactions.

The core equation used by Delft3D-Wave is the spectral action balance equation, which is defined in terms of wave action
density $N(\sigma,\theta)$, where $\sigma$ is the relative frequency, and $\theta$ is the wave direction. The equation is:

$$\frac{\partial N}{\partial t} + \nabla \cdot (\mathbf{c}_g N) + \frac{\partial}{\partial \sigma}(\dot{\sigma}N) + \frac{\partial}{\partial \theta}(\dot{\theta}N) = \frac{S_{in} - S_{out}}{\sigma} \tag{4}$$

where $N = E/\sigma$ is the wave action density (with $E$ being the wave energy density); $\mathbf{c}_g$ is the group velocity of the waves; $\dot{\sigma}$
and $\dot{\theta}$ represent the rate of change in wave frequency and direction, respectively; and $S_{in}$ and $S_{out}$ are source and sink terms

representing processes such as wind input, wave breaking, bottom friction, and whitecapping (Booij et al., 1999).

The coupling between Delft3D-Flow and Delft3D-Wave allows for two-way interaction between hydrodynamic flows and
waves. The wave module provides radiation stress gradients, that are included into the momentum conservation eqauation





(equations 2 and 3), hence considering wave-driven currents and water level variations (wave setup), while the hydrodynamic module provides the water levels and current velocities required by the wave model to compute wave propagation and trans-

190 formation. The coupled system is essential for modeling complex coastal dynamics, where wave-induced currents significantly influence sediment transport and morphological changes, particularly in shallow water environments (Lesser et al., 2004; Holthuijsen, 2010).

### 3.3.2 Model set-up and calibration

The flow model domain was defined as a computational curvilinear grid (blue-Figure 4), with a total of $314 \times 464$ cells and a

195 maximum resolution of $15 \times 25$ m$^2$ within Deception Island. The wave model has two nested grids: a coarse resolution and a fine resolution grids. The fine mesh is defined as above but with more cells, $316 \times 488$ (green-Figure 4), and the coarse mesh has a resolution of $220 \times 160$ m$^2$ with a total of $95 \times 88$ cells (red-Figure 4). Both offshore and Deception Island bathymetry data were provided by the Instituto Hidrográfico de la Marina (Spanish Ministry of Defence) with a resolution of $10 \times 10$ m$^2$. The topography was obtained from Instituto Geográfico Nacional (Spanish Ministry of Transport, Mobility and Urban Agenda)

with a resolution of $5 \times 5$ m$^2$.

The tidal boundary conditions (yellow line-Figure 4) are given by the nine principal astronomical components (semi-diurnal and diurnal constituents) as M2, K1, O1, S2, P1, Q1, K2, N2, and MF. The amplitude and phase of the major tidal constituents from Antarctic Tide Gauge Database (Earth Space Research (ESR), 2024) at the oceanic border have been used. The pressure and 10-meter wind fields required to initialize Delft3D were provided by the WRF simulations, whose description and setup is

205 reported in Section 3.2, providing spatially-variable hourly input data on a 1 km grid. Three wave boundary conditions (purple lines-Figure 4) are defined where the 3-hourly data from the E.U. Copernicus Marine Service reported in Section 3.1.2 was used as input for each one. Furthermore, specific settings used in the Delft3D models are summarized in Table 1.

The simulation period for the calibration runs from December 2007 until March 2008. The model was calibrated using data published in other studies such as (Machado et al., 2011; Antelo et al., 2015; Jigena et al., 2015). The semi-diurnal and diurnal

main harmonic components have been calibrated for the water level and for the currents. Excellent agreement was obtained for water levels and good agreement for currents (Figure 5) with data collected at Cola, Péndulo, Ball and Neptuno stations. These locations are shown in Figure 1, providing a spatial reference for the analysed data sets.

### 4 Data overview

This dataset provides a basis for future research into the hydrodynamics of the island, enabling long-term analyses and com-

215 plementing potential studies that may arise from this work. The integrated approach, involving high-resolution atmospheric downscaling and hydrodynamical modelling, provides a comprehensive view of how Deception Island responds to changing conditions and extreme events, and enhances the understanding of its unique meteo-hydrodynamic processes. The complete atmospheric dataset contains hourly information for 161 atmospheric variables (see Appendix A), including the 10 m wind field and pressure, used as input for the hydrodynamic simulations. The hydrodynamic dataset includes daily data over the whole




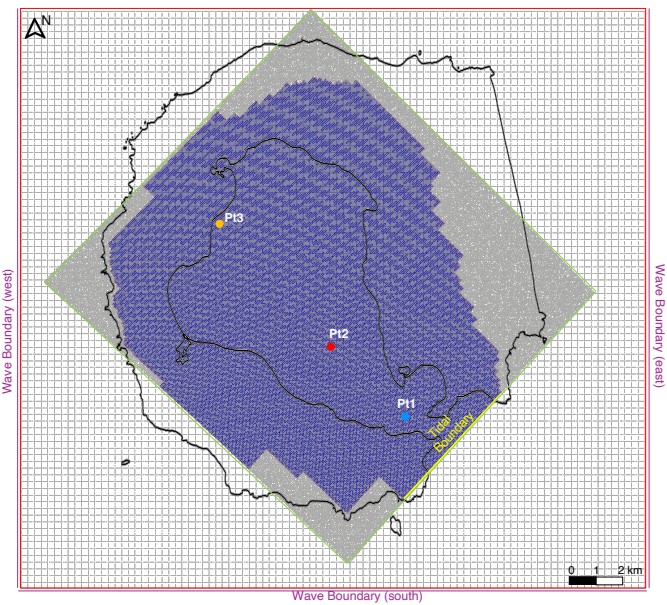

**Figure 4.** Delft3D mesh composition. The red and green polygon correspond to the Delft3D-Wave grids. The blue polygon, covering the island, corresponds to the Delft3D-Flow grid. The yellow line defines the tidal boundary and the purple lines define the three wave boundaries. Wind and pressure fields input are given over the entire grid. The coastline is represented in black color.

| Process | Parameter | Value |
|---|---|---|
| **Flow** | | |
| Time step | - | 0.4 min |
| Bottom roughness | Chezy | Default (65) |
| Stress formulation due to waves | Fredsoe | - |
| Bottom roughness | - | $1 \ \mathrm{m}^2 \cdot \mathrm{s}^{-1}$ |
| Model for 2D turbulence | Deactivated | |
| **Wave** | | |
| Spectral resolution | Directional space | Circle - 36 bins 0-360° |
| | Frequency space | 24 bins 0.05-1 Hz |
| Depth induced breaking | $\alpha$ | 1 |
| | $\gamma$ | 0.73 |
| Nonlinear triad interactions | Deactivated | |
| Diffraction | Deactivated | |
| Bottom friction | JONSWAP | $0.063 \ \mathrm{m}^2 \cdot \mathrm{s}^{-3}$ |
| White capping | Komen | - |

**Table 1.** Parameter descriptions of the flow and wave models.

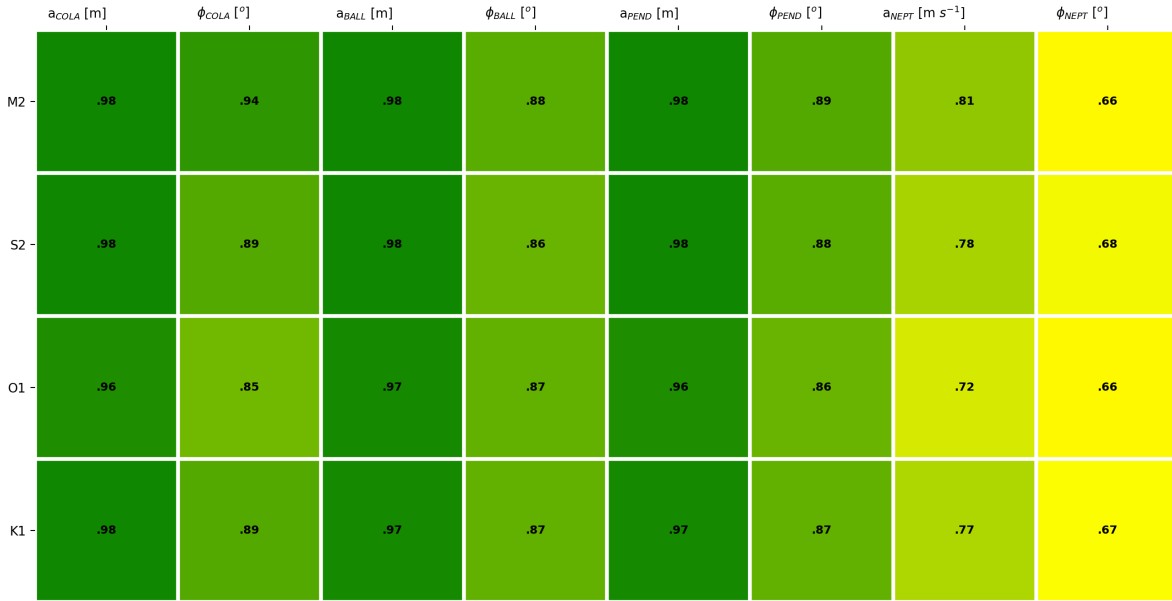

**Figure 5.** Correlation coefficient ($R^2$) of the M2, S2, O1 and K1 constituents of water level and currents for Cola, Pendulo and Ball stations, and Neptuno station, respectively. The color indicates the degree of accuracy (green indicates excellent agreement, yellow-orange indicates good agreement, and red indicates poor agreement).

grid as well as hourly data collected at five observation points strategically distributed over the island. This section presents an analysis of the spatial, seasonal and temporal variability of different key atmospheric and hydrodynamic variables to highlight the capabilities and potential of the combined dataset. In addition, extreme events have been selected to examine their impact on the dynamics of Deception Island.

### 4.1 Data record

The atmospheric dataset comprises, on a ($x = 19$, $y = 20$) Lambert conformal curvilinear grid, hourly information continuously from 2005 until 2022 including a large number of atmospheric variables such as wind velocity and pressure at different vertical levels, precipitation, and temperature, among others. The hydrodynamic dataset covers from 2005 until 2020 although there is some missing data where atmospheric data could not be reproduced or where hydrodynamic simulation errors occurred. The variables recorded include water level, velocity components (east and north), significant wave height, peak period and mean direction. Representative stations with numerical results at high temporal resolution (hourly and daily) were used to analyze the bay dynamics from intratidal to seasonal scales. Figure 6 shows the timeline of the hydrodynamic dataset from 2005 to 2020, highlighting the variables recorded and identifying the gaps. The gaps in the wave boundary data correspond to missing information in the Copernicus dataset; however, to ensure consistency in the dataset, the last available recorded value was kept

constant during data gaps to avoid artificial variability. In contrast, the gaps in the full grid and island point data are due to

computational errors during the hydrodynamic model, resulting in missing records for certain periods.

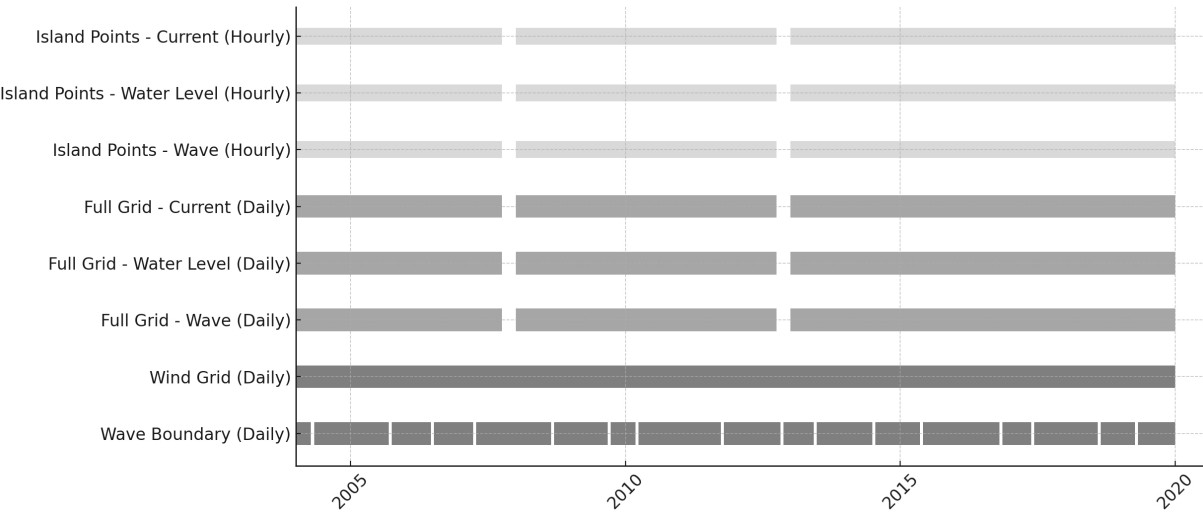

**Figure 6.** Hydrodynamic dataset timeline (2005-2020). The timeline represents different categories of datasets collected over the years. The dataset includes wave boundary, wind grid, full grid (wave, water level and current) and island points (wave, water level and current).

## 4.2    Spatial variability

Figures 7 and 8 present the spatial distribution of the seasonal mean wind and annual mean of the monthly maxima, respectively, over the WRF 1km-domain for the period 2005 until 2022. Regarding the seasonal mean (Figure 7), the highest wind speeds are observed, as expected, during Winter (JJA) with maximum mean wind speed values of over 10 m/s in the Mount Pond,

Whalers bay and Mount Kirkwood regions whereas values of approximately 9 m/s are generally present in Port Foster bay. Mean wind speed values of around 9 m/s are depicted for Spring (SON) and Fall (MAM). The minimum mean speed values are depicted for Summer (DJF) with values of approximate 7 m/s over the entire island. Regarding the spatial variability, all seasons present a similar distribution with the lowest mean wind speeds are depicted in the inner part of Port Forster and the Kendal Terrace whereas the highest mean wind speeds are observed in the mountain areas Goddard Hill, Mount Pond and

Mount Kirkwood as well as in the entrance area, Neptunes Bellows.

Regarding the spatial variability of the annual mean of the monthly maxima wind speed, presented in Figure 8, a similar distribution as the seasonal mean is observed with the highest values depicted for the Mount Pond area and the lowest values observed in the inner Port Foster bay, more specifically, Fumarole Bay. The lowest monthly maxima are depicted for January with wind speed values of approximately 15 m/s whereas the highest values are observed for August with values over 20 m/s.

The spatial variability of key hydrodynamic parameters is shown in Figure 9, comparing conditions during spring and neap tides under both summer and winter scenarios. During spring tides, the highest values of free surface elevation ($\eta$) are observed




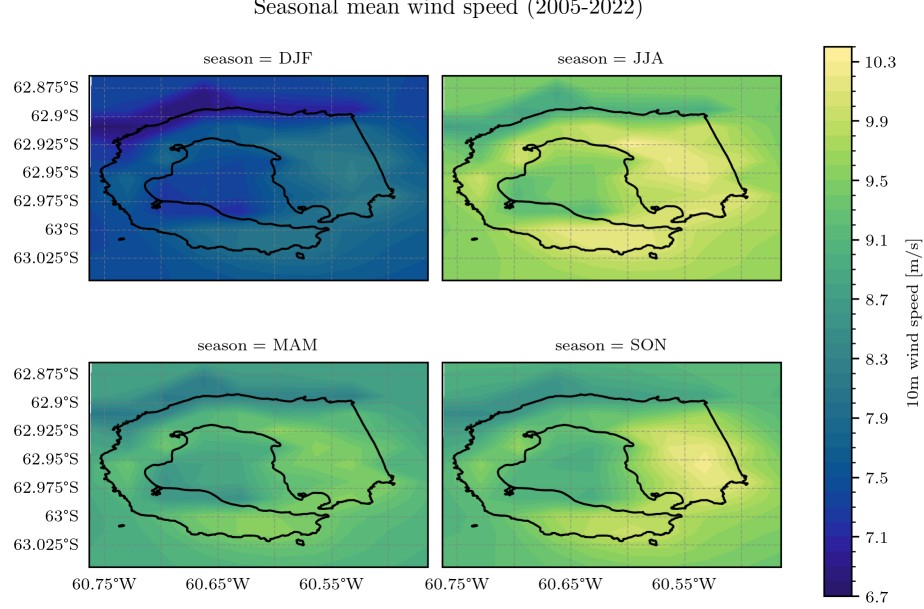

**Figure 7.** Seasonal mean wind speed for the period 2005 until 2022. Coastline data from the SCAR Antarctic Digital Database, 2024

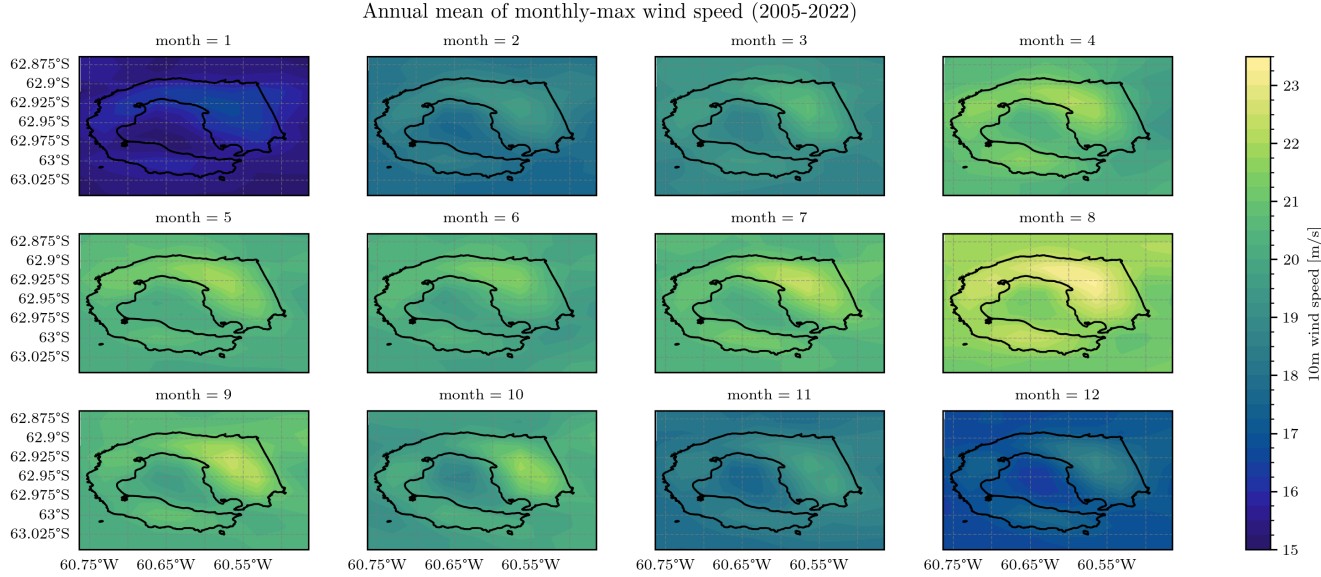

**Figure 8.** Annual mean of monthly maxima wind speed for the period 2005 until 2022. Coastline data from the SCAR Antarctic Digital Database, 2024


**Figure 9.** Panels (a) and (b) display free surface elevation; panels (c) and (d) show velocity; panels (e) and (f) show the significant wave height; and panels (g) and (h) show the wave direction. The number '1' corresponds to summer conditions and '2' to winter conditions. The first and third columns represent spring tides, while the second and fourth columns represent neap tides.

near the coastal boundary, with maximum peaks of about 0.52 m in summer and 0.726 m in winter (panels a.1 and a.2). In contrast, during neap tides, $\eta$ remains generally lower, with peaks around 0.35 m in summer and 0.3 m in winter (panels b.1 and b.2). Current velocity magnitudes follow a similar trend, with higher values during spring tides, reaching up to 0.45 m/s in both seasons (panels c.1 and c.2). During neap tides, velocities are significantly reduced, with maxima below 0.2 m/s (panels d.1 and d.2). Wave parameters ($H_s$ and wave direction) were analysed independently of tidal variability, as wave dynamics are primarily driven by wind forcing and external swell conditions rather than the tidal cycle. The significant wave height ($H_s$) shows seasonal variability, with higher values recorded in winter due to increased storm activity. Maximum values reach 0.4 m in summer (panel e.1) and over 2.5 m in winter (panel e.2). The wave direction shows a spatial gradient with values ranging from -150° to 150° (panels g.1 and g.2).





Seasonal differences between summer and winter conditions (Figure 9, panels 1 and 2) show clear patterns for all analysed parameters. Winter conditions are characterised by significantly higher values of $\eta$, especially during spring tides (panel a.2), and increased significant wave heights, especially in offshore regions where values exceed 2.5 m during neap tides (panel f.2). Summer conditions, on the other hand, exhibit lower $\eta$ and $H_s$, with more uniform spatial distributions across the domain (panels a.1 and e.1). The velocity magnitude shows a slight seasonal increase during the winter spring tides (panel c.2), suggesting stronger hydrodynamic forcing compared to summer (panel c.1). The wave direction also shows seasonal shifts, with a wider directional range in summer (panels g.1, h.1) compared to the more consistent winter patterns (panels g.2, h.2). These results highlight the importance of tidal forcing and seasonality in modulating hydrodynamic and wave dynamics in the study area.

### 4.3 Temporal variability

Figure 10 presents the temporal variability of key atmospheric variables at Neptunes Bellows (location Pt1 - Fig. 4). More specifically, it depicts the hourly time series for wind speed [m/s], wind gusts [m/s], pressure [Pa], temperature [K], relative humidity [%] and precipitation [mm]. Maximum wind speed and wind gusts values of over 20 and 30 m/s are observed with the highest wind speed obtained at the end of 2017. In the pressure panel, the red points represent the annual mean pressure, which does not show any trend over the analyzed years. In the temperature panel, the red and blue dots represent the average summer and winter temperatures, respectively. In this case, an upward trend is clearly visible, particularly during the winter season. Additionally, winter temperatures exhibit greater inter-annual variability compared to summer temperatures.

The time series shown in Figure 11 illustrates the hydrodynamic variability at three observation points along a longitudinal transect in Deception Island: Neptune Bellows (Pt1, blue), the interior of the island in Port Foster (Pt2, red), and a near-shore coastal location in the inner part of the bay (Pt3, orange). The dataset includes water level, current eastward velocity, current northward velocity, significant wave height, peak period and wave direction. The water level shows similar tidal oscillations at all three sites, with slight variations in amplitude. Pt1, being more exposed, shows slightly higher fluctuations compared to the inland and coastal locations where the geography of the island attenuates the tidal signal. The velocity components reflect the dominant tidal influence, with Pt1 showing the highest variations due to exposure to open ocean forcing, while Pt2, located within the island, shows reduced velocity magnitudes, probably due to topographic confinement. Pt3 shows intermediate behaviour, with a balance between tidal and wave driven currents. Significant wave heights follow a similar trend, with Pt1 showing the highest values, occasionally exceeding 1.5 metres during storm conditions. In contrast, wave energy inside the island is significantly reduced, suggesting sheltering effects, while the nearshore point retains moderate variability, influenced by both offshore waves and local conditions. The peak period remains consistent at all locations, but is generally longer at Pt1 due to the influence of offshore swells, while Pt2 mainly reflects shorter periods from locally generated waves. Pt3 shows a mixture of both behaviours, with wave conditions influenced by its semi-exposed position. Wave direction shows greater variability at Pt1, influenced by changing wind and offshore conditions, whereas Pt2 and Pt3 show more stable directional patterns, likely governed by refraction and diffraction effects as waves propagate through the island's channels and coastal topography. Overall, the results highlight a progressive attenuation of hydrodynamic energy from the open ocean towards the sheltered interior,

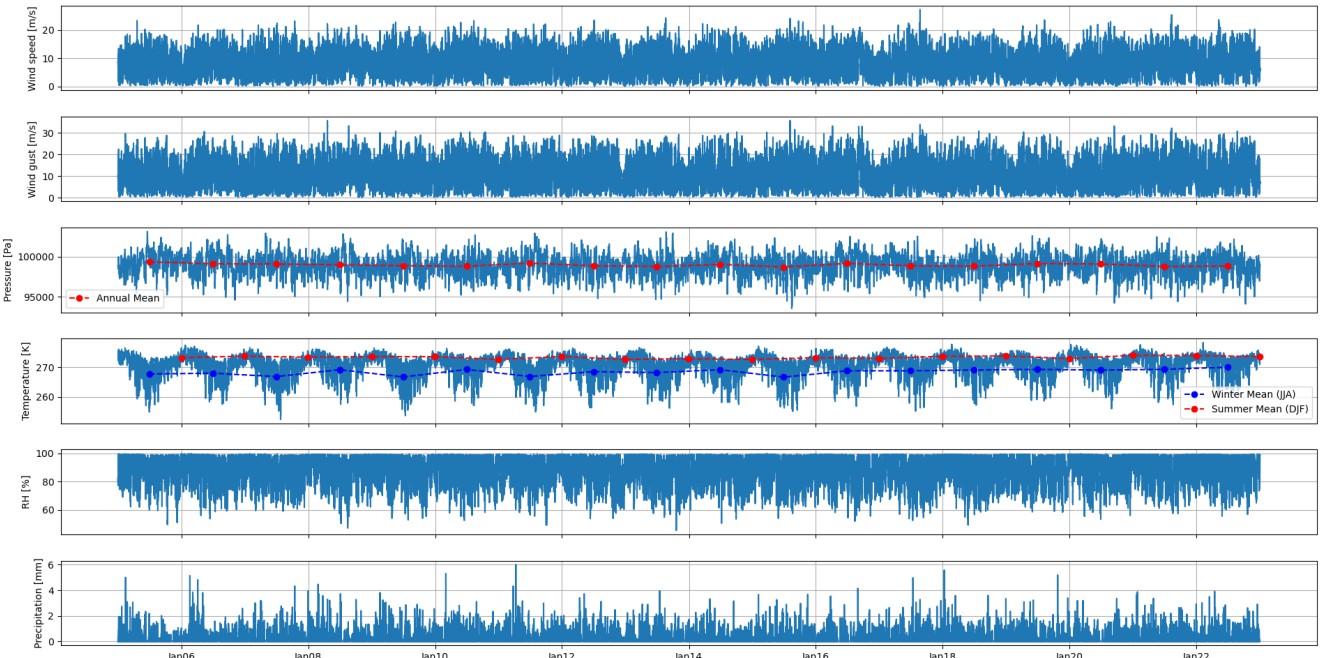

**Figure 10.** Time series (2005-2022) of whether variable for the point Pt1. From top to bottom, variables represented are: wind speed [m/s], wind gust[m/s], pressure [Pa], temperature [K], relative humidity [% ] and hourly precipitation [mm]. Red points in pressure panel refer to pressure annual mean, while red and blue points in temperature panels respectively refer to Summer and Winter mean temperature.

with the nearshore point exhibiting characteristics of both external and internal forcing, highlighting the complex interplay of
tidal currents, wave processes and local bathymetric influences in the hydrodynamic system of Deception Island.

## 4.4    Extreme weather events

Figure 12 shows the temporal evolution of extreme events at the location Pt1 over the analyzed years for sea state (blue), temperature (red), sea level pressure (teal) and wind speed (orange). Concerning sea state (blue line), an event is classified as extreme if the water level or the significant wave height exceeds its 99th percentile for a minimum duration of 2 days. This
threshold ensures that only the most energetic conditions are considered, typically associated with strong storms, high winds or intense oceanic forcing. Regarding weather conditions, temperature and mean sea level pressure extreme events are identified when the daily maximum exceeded the 99th percentile (solid lines) and when the daily minimum was below the 1st percentile (dashed lines) at location Pt1. Regarding wind speed, a location outside the island has been selected and the number of events in which the daily-max exceeded the 99th percentile have been identified.

The variability in the number of extreme events per year reflects fluctuations in atmospheric and hydrodynamic conditions, with certain years experiencing a higher frequency of extreme conditions, possibly related to climatic patterns or seasonal anomalies. Identifying extreme events is crucial to understanding their impact on the hydrodynamics and environmental sta-

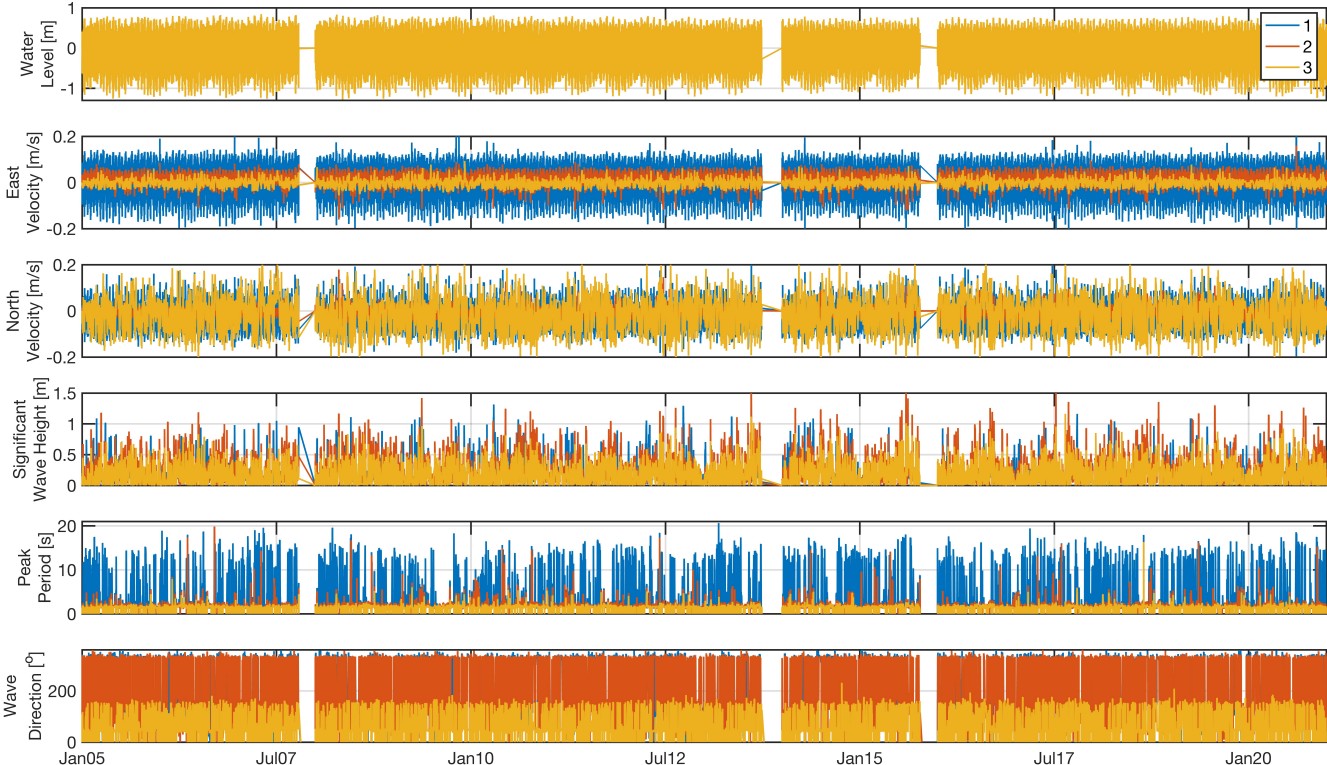

**Figure 11.** Time series of hydrodynamic variables at three observation points aligned along a single longitudinal transect on Deception Island. The blue line corresponds to Neptuno Fuelle, the orange line is located inside the island, and the yellow line is near the coast. From top to bottom, the variables shown are: water level, eastward velocity, northward velocity, significant wave height, peak period, and wave direction.

bility of Deception Island. These events have a significant influence on water level fluctuations, wave dynamics and current patterns, with implications for coastal erosion, sediment transport and potential changes in marine ecosystems.

This period was selected as the most extreme recorded event in the dataset due to its exceptional duration (lasting 7 consecutive days above the 99th percentile) and peak wave heights exceeding 4 metres, making it the most energetic event observed during the study period. Figures 13 and 14 present the behavior of key atmospheric and hydrodynamic variables, during this event. Regarding the atmospheric variables, Figure 13 presents the evolution of wind speed and pressure at Pt2 during the whole event (left) and the spatial distribution (right) in correspondence of the wind peak (12-01-2012 01:00-UTC), and the minimum

pressure during the passage of the storm (14-01-2012 14:00-UTC). Peak wind values of around 15 m/s were identified starting from the 11th of January 2012 and lasting for one day. Afterwards a sudden decrease of up to 2.5 m/s wind speed is observed due to the passage over the island of a pressure minimum associated to the storm. Regarding the spatial distribution during the
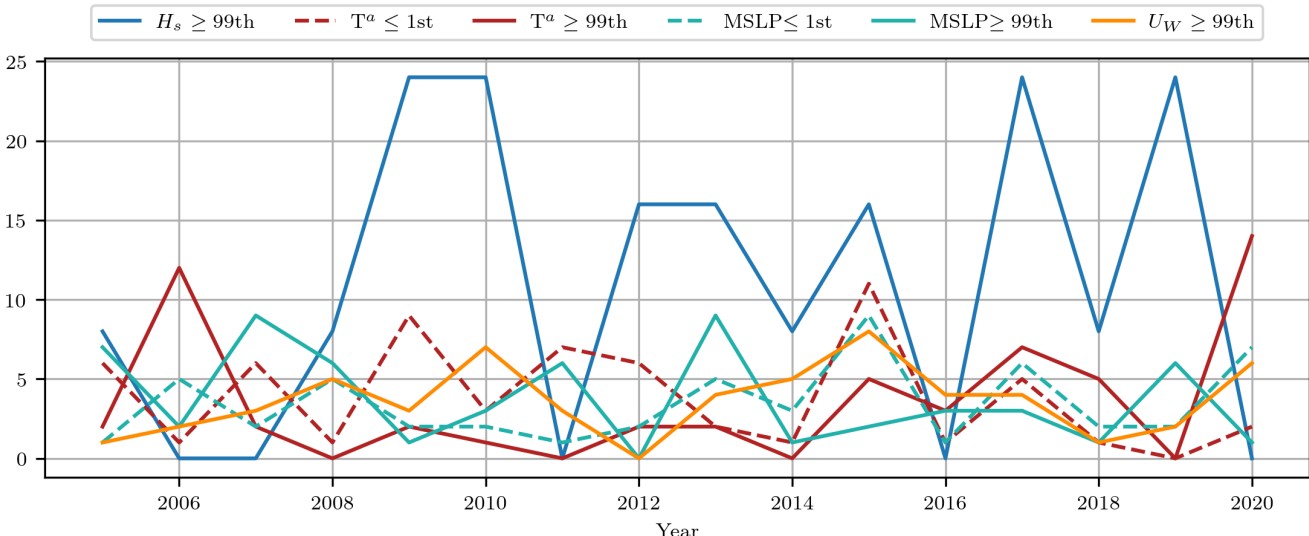

**Figure 12.** Evolution of extreme events over time for different atmospheric and hydrodynamic variables from 2005 to 2020. The blue line represents the number of extreme events, defined as an event where the water level or significant wave height exceeds the 99th percentile. Red and teal lines correspond to the number of events where the daily maxima temperature and mean sea level pressure, respectively, are below the 1st percentile (dashed lines) and above the 99th (solid lines). Orange line corresponds to events where the wind speed exceeds the 99th percentile.

wind speed maxima and pressure minima, it can be observed that the highest values are identified for the mounts' peaks and Neptune Bellows whereas the minima is identified inside the bay in the central part of Port Foster.

Figure 14 shows the spatial distribution of significant wave height (a), peak wave direction (b) and current velocity (c) on the high-resolution Delft3D-Flow grid on the 15th of January 2012. The distribution of the significant wave height ($H_s$), presents maximum values exceeding 1.5 m and predominantly affecting the northern and southeastern regions of Deception Island. This pattern suggests that wave energy is strongly modulated by the island's geomorphology and exposure to external forcing. The wave direction (Fig. 14b) indicates that the waves propagated predominantly from the northwest sector, which is consistent

with the dominant regional wind patterns. However, as the dataset provides daily averages, it is important to recognize that these results represent a single snapshot of the hydrodynamic response of Deception Island during the extreme event, rather than its full temporal evolution.

Finally, Figure 14c shows the current velocity distribution, where the highest values (above 0.6 m/s) are observed along the coastal boundaries, especially near the eastern and northern sectors of the island. This pattern is consistent with the bathymetric

constraints and tidally driven circulation patterns identified in previous Antarctic hydrodynamic modelling efforts Torrecillas et al. (2024). Overall, the strong coupling between wind forcing, wave dynamics and current circulation during extreme events at Deception Island can be observed. The spatial distribution of these variables highlights the role of atmospheric forcing

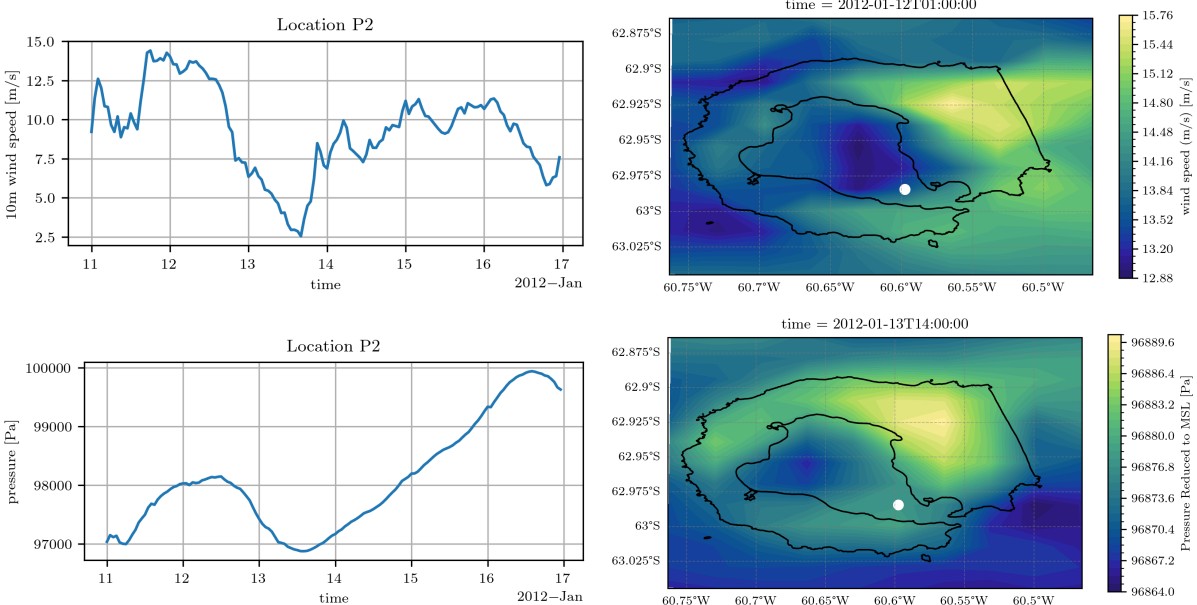

**Figure 13.** Wind speed (top) and pressure (bottom) conditions during an extreme event from the $11^{th}$ until the $17^{th}$ of 15 January, 2012. The left panels present the temporal evolution of the analyzed variables for location Pt2 and the right panels present the spatial distribution for a specific time where high wind speed values and low pressure values were identified. The white dot indicates the location of Pt2. Coastline data from the SCAR Antarctic Digital Database, 2024

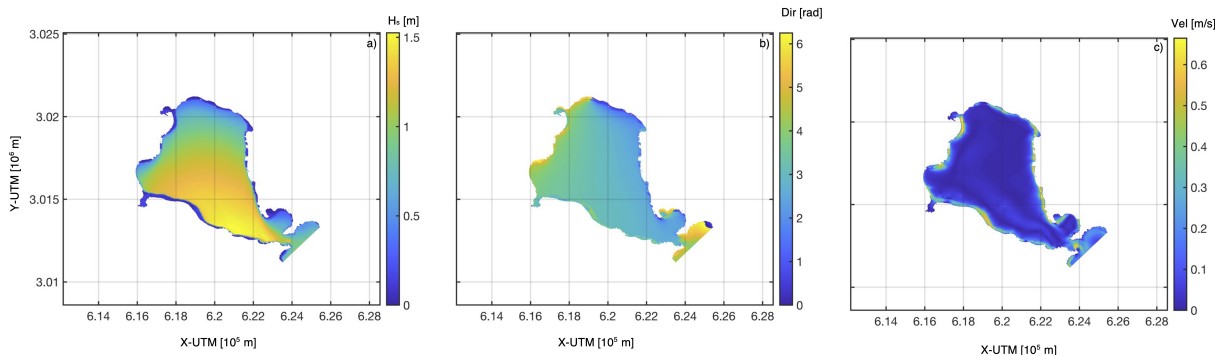

**Figure 14.** Hydrodynamic and atmospheric conditions during an extreme event on January 15, 2012, selected as a case study where the significant wave height exceeded the 99th percentile. Panels show: (a) significant wave height ($H_s$) [m], (b) peak wave direction [radians], and (c) current velocity [m/s]. This analysis provides insight into the spatial variability of wave and wind conditions during extreme events affecting Deception Island.





and island topography in shaping hydrodynamic responses, contributing to a better understanding of the dynamics of extreme events on the island and their potential implications for coastal processes and sediment transport in the region.

## 5 Data availability

The complete atmospheric dataset is stored in yearly NetCDF files for the 1km-SMALL grid (Fig. 2). Each file contains hourly information on a (x=19, y=20) Lambert conformal curvilinear grid for 161 atmospheric variables described in Appendix A (Tables A1-A3), including the 10 m wind field (UGRD_10maboveground and VGRD_10maboveground) and pressure (PRMSL_meansealevel) used as input for the hydrodynamic simulations. The full atmospheric dataset from the WRF simulla- tions comprising all 161 variables can be accessed through the ZENODO open-access data repository, https://doi.org/10.5281/zenodo.14845212 (Ferrari and Loarca, 2025). The dataset employs a consistent file naming as follows: WRF_ANTARTICA_1km_<YY>_SMALL.nc, where <YY> represents each year of the simulation.

The hydrodynamic data presented in this article are freely available at the ZENODO repository. See https://doi.org/10.5281/zenodo.14870881 (Zarzuelo et al., 2025b). The datasets are published in Matlab format (.mat) with the hourly and daily modeled results. The data files (water level, current and wave climate) are explicitly named and contain extensive metadata in the header, indicating whether the data correspond to specific observation points or the full model domain:

- Point-specific hydrodynamic model data: *Hydrodynamics_PtX*, where $X$ represents the station number

- Full hydrodynamic model domain: *data20XX*, where $20XX$ represents the simulation year

## 6 Conclusions

A unique and comprehensive dataset has been developed for Deception Island, Antarctica, combining the WRF atmospheric model and a hydrodynamic model forced using the FLOW and WAVE modules. The resulting atmospheric dataset includes 161 atmospheric variables such as precipitation, temperature, pressure at mean sea level, wind velocity, and relative humidity, among other. The hydrodynamic datasets includes water level, current velocity, significant wave height, mean wave period and wave direction, providing high spatial and temporal resolution data that capture the complex dynamics of the island's environment, including extreme events.

These datasets allow detailed analysis of the spatial, seasonal and temporal variability of the island's hydrodynamics. It also provides insights into the impact of extreme events on these dynamics. As shown in the analyses, the dataset facilitates: (1) the study of the spatial variability of key atmospheric (wind speed, pressure) and hydrodynamic (water level and velocity) variables across the island, (2) their seasonal variability, and (3) the influence of extreme atmospheric conditions, such as intense wind events, on hydrodynamic variables.

Furthermore, these datasets can be used for various applications, such as investigating the influence of wind and waves on hydrodynamic processes, or assessing the interactions between physical and ecological dynamics on the island. The data can



also be used to support long-term environmental monitoring and model development to better understand the potential impacts of future climate change, such as sea level rise or shifts in storm patterns, on the fragile Antarctic environment.



## Appendix A: WRF data variables

| Variable | Long Name | Units | Levels |
|---|---|---|---|
| PRMSL_meansealevel | Pressure Reduced to MSL | Pa | mean sea level |
| MSLET_meansealevel | MSLP (Eta model reduction) | Pa | Mean sea level |
| MSLMA_meansealevel | MSLP (MAPS System Reduction) | Pa | Mean sea level |
| PRES_surface | Pressure | Pa | Surface |
| HGT_surface | Geopotential Height | m | Surface |
| TMP_surface | Temperature | K | Surface |
| WEASD_surface | Water Equivalent of Accumulated Snow Depth | $kg/m^2$ | Surface |
| SNOWC_surface | Snow Cover | Percent | Surface |
| SNOD_surface | Snow Depth | m | Surface |
| PRATE_surface | Precipitation Rate | $kg/m^2/s$ | Surface |
| APCP_surface | Total Precipitation | $kg/m^2$ | Surface |
| ACPCP_surface | Convective Precipitation | $kg/m^2$ | Surface |
| NCPCP_surface | Large-Scale Precipitation (non-convective) | $kg/m^2$ | Surface |
| PEVAP_surface | Potential Evaporation | $kg/m^2$ | Surface |
| SFCR_surface | Surface Roughness | m | Surface |
| FRICV_surface | Frictional Velocity | m/s | Surface |
| SHTFL_surface | Sensible Heat Net Flux | $W/m^2$ | Surface |
| LHTFL_surface | Latent Heat Net Flux | $W/m^2$ | Surface |
| CAPE_surface | Convective Available Potential Energy | J/kg | Surface |
| CIN_surface | Convective Inhibition | J/kg | Surface |
| HPBL_surface | Planetary Boundary Layer Height | m | Surface |
| NLAT_surface | Latitude (-90 to 90) | deg | Surface |
| ELON_surface | East Longitude (0 to 360) | deg | Surface |
| LAND_surface | Land Cover (0=sea, 1=land) | Proportion | Surface |
| ICEC_surface | Ice Cover | Proportion | Surface |
| WTMP_surface | Water Temperature | K | Surface |
| VIS_surface | Visibility | m | surface |
| GUST_surface | Wind Speed (Gust) | m/s | surface |
| HGT_$P$mb | Geopotential Height | m | $P$ mb<br>$P$: 200, 300, 500, 700, 800, 850, 875, 900, 925, 950, 975, 1000 |

**Table A1.** Description of atmospheric variables from the WRF simulations





| Variable | Long Name | Units | Levels |
|---|---|---|---|
| TMP_Pmb | Temperature | K | P mb<br>P: 200, 300, 500, 700, 800, 850, 875, 900, 925, 950, 975, 1000 |
| SPFH_Pmb | Specific Humidity | kg/kg | P mb<br>P: 200, 300, 500, 700, 800, 850, 875, 900, 925, 950, 975, 1000 |
| VVEL_Pmb | Vertical Velocity (Pressure) | Pa/s | P mb<br>P: 200, 300, 500, 700, 800, 850 |
| UGRD_Pmb | U-Component of Wind | m/s | P mb<br>P: 200, 300, 500, 700, 800, 850, 875, 900, 925, 950, 975, 1000 |
| VGRD_Pmb | V-Component of Wind | m/s | P mb<br>P: 200, 300, 500, 700, 800, 850, 875, 900, 925, 950, 975, 1000 |
| ABSV_Pmb | Absolute Vorticity | 1/s | P mb<br>P: 200, 300, 500, 700, 800, 850 |
| CLWMR_Pmb | Cloud Mixing Ratio | kg/kg | P mb<br>P: 200, 300, 500, 700, 800, 850, 875, 900, 925, 950, 975, 1000 |
| CICE_Pmb | Cloud Ice | kg/m$^2$ | P mb<br>P: 200, 300, 500, 700, 800, 850, 875, 900, 925, 950, 975, 1000 |
| TMP_2maboveground | Temperature | K | 2 m above ground |
| SPFH_2maboveground | Specific Humidity | kg/kg | 2 m above ground |
| DPT_2maboveground | Dew Point Temperature | K | 2 m above ground |
| RH_2maboveground | Relative Humidity | % | 2 m above ground |
| UGRD_Lmaboveground | U-Component of Wind | m/s | L m above ground<br>L: 10, 20, 30, 40, 50, 80 |
| VGRD_Lmaboveground | V-Component of Wind | m/s | L m above ground<br>L: 10, 20, 30, 40, 50, 80 |
| SNOHF_10maboveground | Snow Phase Change Heat Flux | W/m$^2$ | 10 m above ground |
| REFD_4000maboveground | Reflectivity | dB | 4000 m above ground |
| UGRD_30M0mbaboveground | U-Component of Wind | m/s | 30-0 mb above ground |
| VGRD_30M0mbaboveground | V-Component of Wind | m/s | 30-0 mb above ground |
| SPFH_30M0mbaboveground | Specific Humidity | kg/kg | 30-0 mb above ground |
| HGT_0Cisotherm | Geopotential Height<br>(0C Isotherm) | m | 0C Isotherm |

**Table A2.** Continued: Description of atmospheric variables from the WRF simulations



| Variable | Long Name | Units | Levels |
|---|---|---|---|
| REFC_*P* | Composite reflectivity | dB | Entire Atmosphere (Single Layer) |
| | | | *P*: entireatmosphere_consideredasasinglelayer |
| PWAT_*P* | Precipitable Water | kg/m$^2$ | Entire Atmosphere (Single Layer) |
| | | | *P*: entireatmosphere_consideredasasinglelayer |
| TCOLW_*P* | Total Column-Integrated Cloud Water | kg/m$^2$ | Entire Atmosphere (Single Layer) |
| | | | *P*: entireatmosphere_consideredasasinglelayer_ |
| TCOLI_*P* | Total Column-Integrated Cloud Ice | kg/m$^2$ | Entire Atmosphere (Single Layer) |
| | | | *P*: entireatmosphere_consideredasasinglelayer_ |
| TCOLR_*P* | Total Column Integrated Rain | kg/m$^2$ | Entire Atmosphere (Single Layer) |
| | | | *P*: entireatmosphere_consideredasasinglelayer_ |
| TCOLS_*P* | Total Column Integrated Snow | kg/m$^2$ | Entire Atmosphere (Single Layer) |
| | | | *P*: entireatmosphere_consideredasasinglelayer_ |
| TCOLC_*P* | Total Column-Integrated Condensate | kg/m$^2$ | Entire Atmosphere (Single Layer) |
| | | | *P*: entireatmosphere_consideredasasinglelayer_ |
| TCDC_*P* | Total Cloud Cover | Percent | Entire Atmosphere (Single Layer) |
| | | | *P*: entireatmosphere_consideredasasinglelayer_ |
| LCDC_lowcloudlayer | Low Cloud Cover | Percent | Low Cloud Layer |
| MCDC_middlecloudlayer | Medium Cloud Cover | Percent | Middle Cloud Layer |
| HCDC_highcloudlayer | High Cloud Cover | Percent | High Cloud Layer |
| HGT_cloudbase | Geopotential Height (Cloud Base) | m | Cloud Base |
| HGT_cloudceiling | Geopotential Height (Cloud Ceiling) | m | Cloud Ceiling |
| HGT_cloudtop | Geopotential Height (Cloud Top) | m | Cloud Top |
| HGT_tropopause | Geopotential Height (Tropopause) | m | Tropopause |

**Table A3.** Continued: Description of atmospheric variables from the WRF simulations

*Author contributions.* C.Z, A.L.R., F.F and A.L.L conceptualized the study and the manuscript, performed data analysis and visualization and prepared the original draft of the manuscript as well as reviewed and edited the final manuscript. F.F. set up the WRF atmospheric numerical model and performed the WRF downscaling simulations, A.L.R. and C.Z. set up, calibrated and validated the DELFT3D hydrodynamics numerical model, performed the hydrodynamic simulations and prepared the hydrodynamics dataset, A.L.L performed the validation of the
370 WRF model, prepared the atmospheric and wave input data, contributed to the DELFT3D model set up and prepared the atmospheric dataset.

*Competing interests.* The authors declare no competing interests.



*Acknowledgements.* This work has been supported by the Spanish Ministry of of Economy and Competitiveness, PID2021-125895OA-I00 (RESILIENCE), and by Department of Economy, Knowledge, Business and Universities of the Andalusian Regional Government (Project A-TEP-88-UGR20).





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
