# Peer review of "Hydrodynamic and Atmospheric Conditions in a Volcanic Caldera: A Comprehensive Dataset at Deception Island, Antarctica"

_Earth System Science Data, 2025_

## Author Response (AR1)

**Comment: Hydrodynamic and Atmospheric Conditions in a Volcanic Caldera: A Comprehensive Dataset at Deception Island, Antarctica**

This study presents the first 16-year (2005–2020) high-resolution integrated atmospheric-hydrodynamic dataset for Deception Island, a volcanic caldera in Antarctica, combining nested WRF atmospheric simulations (1 km grid) and DELFT3D hydrodynamic modeling (15×25 m grid). The dataset addresses critical knowledge gaps in Antarctic coastal systems by capturing spatial-seasonal variability and extreme events (e.g., storm-driven waves >4 m).

The integration of WRF downscaling (validated against in-situ weather stations) and DELFT3D (calibrated with tidal harmonics) enables robust analysis of wind-wave-current interactions during extreme events. The dataset supports diverse applications, including glacial meltwater transport, nutrient dynamics, and ecosystem resilience, aligning with Antarctic conservation priorities.

1. The correlation coefficients for temperature (R=0.86–0.99) and pressure (R=0.99) are robust, but lower correlations for precipitation (R=0.43) and humidity (R=0.57) require deeper analysis.

Precipitation is linked to the micro-physics parameterization and both a long and accurate observation time-series and an in-depth sensitivity analysis in polar areas, would be needed to accurately calibrate model precipitation. In fact, precipitation in areas characterized by polar climate could need the development and/or the specific validation of the parameterization scheme to improve reliability in precipitation simulations.  Furthermore, for high-resolution simulations such, punctual comparison between observation and forecast is affected by the so called -double penalty- effect, that result in degradation of traditional standard verfication index (Cassola *et al.*, 2015). A more sophisticated analysis could required, for example, an object oriented validation of the simulations, based on the direct comparison between observed and forecast precipitation fields (Ferrari *et al.*, 2023). Likewise, humidity can partially be influenced by precipitation leading to lower correlations. Additionally, the reliability of observation of solid precipitation, like snow, could be strongly affected by wind and with this region being characterized by both strong winds and solid precipitation, inaccuracies between observations and model could be detected.  For this, lower correlations are expected for variables such as precipitation and humidity than temperature or pressure. Nonetheless, considering the 1km grid for the WRF model, we believe the correlations obtained for all variables are adequate to use for future analysis. This has been further clarified in the revised version of the manuscript.

Cassola, F., Ferrari, F. and Mazzino, A., 2015. Numerical simulations of Mediterranean heavy precipitation events with the WRF model: A verification exercise using different approaches. *Atmospheric Researc*h, 164, pp.210-225.
Ferrari, F., Maggioni, E., Perotto, A., Salerno, R. and Giudici, M., 2023. Cascade sensitivity tests to model deep convective systems in complex orography with WRF. *Atmospheric Research*, *295*, p.106964.

1. Tidal constituents (M2, S2) show excellent agreement with historical data, yet the absence of satellite altimetry validation limits confidence in open-ocean boundary conditions.

We thank the reviewer for this valuable comment. Unfortunately, historical satellite altimetry data with sufficient spatial and temporal resolution are not available for Deception Island, which prevents us from using such data for model calibration. However, our study focuses primarily on the hydrodynamic behaviour within the island's caldera rather than in the open ocean. Model calibration was performed using in-situ observations from previously published studies, providing a robust validation of the numerical model for our area of interest. Furthermore, even if satellite altimetry with a sufficient resolution was available, it would require additional calibration and validation against in-situ measurements to be fully reliable in this particular environment. We have clarified this point in the revised manuscript.

1. Missing hydrodynamic data (2005–2020) due to computational errors or Copernicus boundary gaps need explicit justification. The 10×10 m bathymetry may inadequately resolve narrow channels. Higher-resolution terrain data should be tested to assess sensitivity.

We appreciate the reviewer's comment regarding the missing hydrodynamic data and the resolution of the bathymetric input. Regarding the missing hydrodynamic data during the 2005–2020 period, the gaps are mainly due to occasional computational instabilities in the numerical hydrodynamic model and missing input data from the Copernicus Marine Service used as boundary conditions. To prevent introducing artificial variability, simulations were halted during such periods, and no gap-filling or synthetic reconstruction was applied, preserving the physical consistency and reliability of the dataset. Regarding the bathymetric resolution, although 10×10 m data were used, our study's focus is on basin-scale atmospheric and hydrodynamic variability, not on detailed geomorphological or fine-scale channel flows. Therefore, the current resolution is considered appropriate for the study objectives. Nevertheless, we acknowledge the importance of high-resolution bathymetric data for studies specifically targeting channel dynamics, and we have included this limitation in the revised manuscript.

*"The missing hydrodynamic data for certain periods between 2005 and 2020 are mainly due to computational instabilities or the unavailability of boundary condition data from the Copernicus Marine Service. No artificial gap-filling has been applied to maintain the physical reliability of the dataset. Although the bathymetric data used have a horizontal resolution of 10×10 m, this is considered sufficient for the study objectives, which focus on basin-scale hydrodynamic and atmospheric variability. Future studies specifically targeting detailed channel dynamics may benefit from the use of higher resolution terrain data."*

1. The seasonal wind speed maxima (>10 m/s in winter) and wave height contrasts (summer: 0.4 m vs. winter: 2.5 m) are well-documented. However, linking these trends to broader climate signals would enhance relevance.

We thank the reviewer for the insightful suggestion to link observed seasonal variability to broader climatic signals. In response, we have expanded the manuscript by explicitly discussing the relationship between local wind speed and large-scale climate variability, specifically the Southern Annular Mode (SAM) and the El Niño–Southern Oscillation (ENSO). We have added a new paragraph in the results and discussion section, complemented by a new figure (Figure X) that compares the monthly mean wind speed at Deception Island with the SAM index over the period 2005–2020. This addition highlights the influence of positive SAM phases on stronger wind conditions observed locally. Furthermore, we briefly discuss the potential effects of ENSO phases on atmospheric and hydrodynamic variability in the region, suggesting directions for future research. We believe these additions strengthen the broader climatological relevance of our work and directly address the reviewer's recommendation.

[Figure]

*"Figure X.-Monthly wind speed (blue line) and Southern Annular Mode (SAM) index (red dashed line) at Deception Island from 2005 to 2020, illustrating the relationship between local wind variability and large-scale climate patterns.*

*The observed seasonal variability in wind speed and wave height at Deception Island, characterized by higher values during austral winter, is consistent with broader climate patterns affecting the Southern Ocean. In particular, phases of the Southern Annular Mode (SAM) are known to modulate the intensity and persistence of westerly winds over the South Shetland Islands region (e.g., Marshall, 2003). Figure X shows the comparison between the monthly mean wind speed at Deception Island and the Southern Annular Mode (SAM) index over the period 2005-2020. The SAM index describes the difference in zonal mean sea level pressure between approximately 40°S and 65°S, capturing the variability of the westerly wind belt that surrounds Antarctica. Positive SAM phases are associated with stronger and poleward shifting westerlies, while negative phases indicate weakening and equatorward shifting. The figure shows a general correspondence between periods of positive SAM and higher wind speeds at Deception Island, suggesting that local atmospheric conditions are modulated, at least in part, by broader climatic*

*patterns. This relationship highlights the potential influence of large-scale climate variability on the hydrodynamic and atmospheric processes studied in this work.*

*In addition to the Southern Annular Mode (SAM), the El Niño-Southern Oscillation (ENSO) also influences atmospheric and oceanographic conditions in the Antarctic Peninsula region. El Niño events are generally associated with weaker westerly winds and reduced storm activity, while La Niña phases tend to increase wind strength and hydrodynamic forcing. Although the present analysis focuses primarily on local variability, future studies could explore the relationship between ENSO phases and the patterns observed at Deception Island."*

1. The storm case study highlights wave-current coupling but lacks analysis of climate-driven frequency changes.

We thank the reviewer for this important observation. The storm case study was selected to illustrate the coupled dynamics of wind, waves, and currents during an extreme event. While a comprehensive trend analysis of storm frequency changes was beyond the scope of the present work, we agree that investigating potential climate-driven changes in the occurrence of such events is highly relevant. In response, we have added a discussion point acknowledging this limitation and suggesting that future studies could integrate climate indices and longer-term trend analyses to assess changes in the frequency and intensity of extreme hydrodynamic conditions in the region. This would enhance the predictive power of the dataset in the context of ongoing climate change.

*"While this case study illustrates the dynamic response of the system to an extreme weather event, the dataset also provides an opportunity for future analyses on the frequency and intensity of such events in relation to large-scale climate variability. Investigating long-term trends and their potential links to climate change, including the role of SAM and ENSO, could offer valuable insights into the evolving risk of extreme hydrodynamic conditions in the Antarctic Peninsula region."*

1. Incorporate satellite altimetry or Argo float data to validate hydrodynamic outputs in open-ocean regions.

We appreciate the reviewer's suggestion to incorporate satellite altimetry or Argo float data to validate hydrodynamic outputs in open-ocean regions. However, due to the geographic and environmental constraints of the Southern Ocean, such high-resolution data are either scarce or unavailable for the Deception Island region during the simulation period (2005–2020). Specifically, access to Argo float data for this region and period reveals that measurements are irregular both spatially and temporally, with significant data gaps, especially under ice cover. Moreover, the available Argo data, when accessible, refer to undefined depths, and are not suitable for surface water level validation; similarly, attempts to extract pressure-related values at the necessary resolution and time range (2005–2020) have been unsuccessful.

Additionally, our study focuses on the internal hydrodynamic processes within the Deception Island caldera rather than open-ocean dynamics, making these external datasets less relevant to our specific research objectives.

Furthermore, we are not specialists in the processing and interpretation of satellite altimetry or Argo float data, and using such datasets without appropriate expertise could introduce additional sources of uncertainty. Our calibration and validation approach, based on in-situ observational data from previous published studies, provides a reliable and robust basis for assessing model performance within the scope of this work. This point has been clarified in the revised manuscript.

This study delivers a pioneering dataset for Antarctic volcanic caldera systems, with significant potential for cross-disciplinary research. Addressing validation gaps and climate linkages will solidify its impact.

The dataset proposed by Ferrari et al. provides spatially resolved fields of atmospheric and hydrodynamic variables for the caldera region within Deception Island, based on a long-term one-way coupled numerical experiment. I appreciate the clear presentation of the dataset, the quality of the figures, and the logical flow of the manuscript. Furthermore, I find it commendable that the authors dedicate significant attention to sub-scaling atmospheric models, nesting multiple grids and models, and rigorously calibrating and validating various variables across different spatial and temporal scales. These are efforts often overlooked in research that is primarily question-driven, yet they deserve recognition in their own right. This dataset holds considerable potential for future applications, particularly in the validation and comparison of remote sensing products. As sensor resolution continues to improve, it could become increasingly valuable for comparisons with water levels from radar altimetry and SAR-derived wind, waves and currents fields.

I have a few minor comments I'd like the authors to address to improve the manuscript's clarity and enhance the reader's understanding of the modelling framework and decisions.

WRF is defined 3 times: L25, 30, 116. One is enough

*This has been corrected and WRF has been defined once when it first appears.*

Figure 1 and lines 69–72: The bathymetry inside the caldera is a bit under-described. While the narrow passage depth is mentioned, the central caldera basin appears to exceed 100 m depth based on the figure, yet this is not stated in the text. Please include this detail in the description of the case study and consider refining the color classification in Figure 1 to better represent bathymetric variation.

[Figure]

*"The inner basin of Port Foster reaches depths exceeding 100 m, with some areas approaching 180 m, as also depicted in Figure 1. This relatively deep and enclosed environment plays a key role in modulating internal circulation and wave attenuation."*

Section 3.2.1: maybe I missed it, and that's just a curiosity: what is the resolution of the digital elevation model for the island used in the WRF domain for downscaling? Did I get it right that WRF was not specifically calibrated for this application but only validated?

For the WRF atmospheric simulations, the spatial resolution of the static data (orography, land use, and soil type) is 30 arc seconds, which corresponds to approximately 1 km coherently with the atmospheric simulation resolution.  WRF was validated and not specifically calibrated, although specific polar settings were used as described in the methodology. Specific calibration would have required sensitivity analysis to test multiple parameterisations of different physical phenomena (microphisycs, radiation, PBL, ...) resulting in an very high computational effort.

Figure 2: Does "highlighted inlet" refer to the white box in the left panel? Is the color scale depicting topography? Please clarify this in the caption and add a label to the color bar.
Yes, the highlighted inlet refers to the white box and the zoom-figure on the right. This has been clarified in the caption as well as the fact that the color scale depicts topography in meters.

Section 3.3.1: I have some points regarding the hydrdodynamic modelling choices:
1. please explicitly state here and in the abstract that the Delft3D model chosen is in its 2D version (if I understood it correctly). Given the depth of the caldera I would have honestly expected a 3D model. Maybe the authors can comment and justify this choice here?

Yes, the hydrodynamic simulations presented in this study were performed using the 2D depth-averaged version of Delft3D-FLOW. This choice was driven primarily by the lack of sufficient vertical observational data needed to calibrate and validate a full 3D hydrodynamic model. Specifically, the absence of vertical profiles of temperature, salinity, and current velocity, as well as limited knowledge of key calibration parameters such as vertical turbulent viscosity and diffusivity coefficients, made a 3D configuration unfeasible within the current scope. We have now explicitly stated the use of the 2D model in both the abstract and the main text, and clarified the rationale for this decision.

*"Delft3D is a widely used numerical modeling system for simulating hydrodynamic, sediment transport, and morphodynamic processes in coastal, estuarine, and riverine environments. Developed by Deltares, the model consists of several interlinked modules, each capable of handling specific physical processes. Among them, modules Flow and Wave were used in this work to obtain wind, pressure, tidal and wave generated hydrodynamics. In this study, the depth-averaged version of the flow was applied to simulate the hydrodynamic conditions at Deception Island."*

1. Am I understanding correctly that the authors did not implement the heat fluxes module and did not simulate temperature? I wonder why since they would have had everything they needed from WRF. Maybe no in-situ temperature data for validation?

Yes, the reviewer is correct — the heat flux module was not implemented, and temperature was not simulated in the hydrodynamic model. Although surface atmospheric forcing (e.g.,

radiation, air temperature, wind) from WRF was available, the decision to not activate the heat flux module was due to the lack of in-situ water temperature data required for reliable calibration and validation. Implementing thermal dynamics without appropriate validation could have introduced additional uncertainties.

1. Was the model calibration carried out via trial and error, or was an automated method employed? Please specify. Please specify how modeled currents were compared with in-situ measurements. I understand most information is contained in the references to Machado et al., 2011; Antelo et al., 2015; Jigena et al., 2015, but something should be also clarified here. Are in-situ currents measured at the surface, deeper points, depth-averaged? Any idea why such a different performance is obtained in Neptuno station compared to the others? Maybe local effects not correctly represented? Any influence of 3D processes not accounted for in the 2D simulation? Future users must be aware of what these model results are representative or not for. See also my comment to section 4.2 and 4.3.

The model calibration was carried out using a manual trial-and-error approach, as is common in coastal environments with limited observational data. The calibration period extended from December 2007 to March 2008, and relied on in-situ data published in Machado et al. (2011), Antelo et al. (2015), and Jigena et al. (2015).

These previous studies provided tidal harmonic components for water level and depth-averaged current harmonics, which were used to calibrate the dominant semi-diurnal (M2, S2) and diurnal (K1, O1) tidal constituents. The model showed excellent agreement for water level and good agreement for currents, as shown in Figure 9, with validation at four stations: Cola, Péndulo, Ball, and Neptuno (see Figure 1 for location).

The lower performance observed at Neptuno station likely reflects localised effects such as complex bathymetry, abrupt coastline features, and geoide-induced variations. In that area, even small spatial displacements result in substantial changes in current speed and direction. Additionally, the use of a depth-averaged (2DH) model does not capture potential 3D processes (e.g., vertical shear, stratification, density-driven flows), which may play a relevant role in such a confined and dynamic location. These limitations have now been clarified in the manuscript to help future users understand the representativeness and appropriate interpretation of the dataset.

*"The model calibration was carried out using a manual trial-and-error approach, as is common in coastal regions where data availability is limited. The calibration period extended from December 2007 to March 2008, and relied on in-situ data published in previous studies by Machado et al. (2011), Antelo et al. (2015), and Jigena et al. (2015). These datasets provide tidal harmonic components for water level and depth-averaged current harmonics. The calibration focused on the dominant semi-diurnal (M2, S2) and diurnal (K1, O1) tidal constituents, both for water level and for currents. As described in the manuscript (see Figure \ref{fig:cal}), excellent agreement was achieved for water levels, and good agreement was obtained for currents at four monitoring sites: Cola, Péndulo, Ball, and Neptuno, which are spatially distributed across the island (see Figure \ref{f1}). The lower model performance observed at Neptuno station may*

*reflect localised hydrodynamic effects not well resolved in the 2D simulation. This includes steep bathymetric gradients, coastline irregularities, and potentially 3D circulation patterns (e.g., density-driven flows or vertical shear) that are not captured in a depth-averaged model. These limitations have been clarified to inform future users about the representativeness and appropriate use of the model results, especially in areas with complex nearshore dynamics."*

Figure 4 Please label the observation stations (Cola, Péndulo, Ball, Neptuno) directly in the figure as they are referred to in the text. Additionally, the green polygon is hard to distinguish—consider using a more contrasting color.

[Figure]

Table 1: Add units for the Chezy coefficient. Also, the type of coupling used between FLOW and SWAN should be specified. Although the authors discuss two-way coupling in lines 186–193, the

actual choice made is not clearly stated. Delft3D-SWAN allows several coupling options (e.g., "only use", "use and extend", etc.); including this in the table would benefit other modelers referencing this work.

We have added the unit of the Chezy roughness coefficient ($m^{1/2} \cdot s^{-1}$) to Table 1 for clarity. Additionally, we have clarified the type of coupling used between FLOW and SWAN in the Delft3D model. Specifically, the "use and extend" option was applied, allowing wave effects to be included in the hydrodynamic module.

| Process | Parameter | Value |
|---|---|---|
| **Flow** | | |
| Time step | - | 0.4 min |
| Bottom roughness | Chezy | Default ($65\ m^{1/2} \cdot s^{-1}$) |
| Stress formulation due to waves | Fredsoe | - |
| Bottom roughness | - | $1\ m^2 \cdot s^{-1}$ |
| Model for 2D turbulence | Deactivated | |
| **Wave** | | |
| Spectral resolution | Directional space | Circle - 36 bins 0-360° |
| | Frequency space | 24 bins 0.05-1 Hz |
| Depth induced breaking | $\alpha$ | 1 |
| | $\gamma$ | 0.73 |
| Nonlinear triad interactions | Deactivated | |
| Diffraction | Deactivated | |
| Bottom friction | JONSWAP | $0.063\ m^2 \cdot s^{-3}$ |
| White capping | Komen | - |
| Flow-Wave coupling | Use and extend | - |

**Table 1.** Parameter descriptions of the flow and wave models.

Figure 6: I don't get the colour code in figure 6, why are some grey bars lighter than the others? Is it just an aesthetical choice? If yes, I suggest to keep the same colour to avoid confusion.

The different grey tones were used intentionally to distinguish between the temporal resolution and spatial coverage of the data: light grey represents hourly point data, medium grey indicates daily full grid, and dark grey corresponds to full grid data. In addition, we have updated the figure to represent the wave boundary and wind forcing datasets in black, as suggested, to avoid confusion and clearly separate these inputs from the hydrodynamic outputs. We have also revised the figure caption to explicitly explain the color scheme for clarity.

*"Hydrodynamic dataset timeline (2005–2020). The figure shows the availability of different components of the dataset across time. Light grey bars represent hourly point data (wave, water level, and current), while darker grey bars indicate daily point data. The darkest bars*

*correspond to data available over the full model grid. Color shading indicates the temporal resolution and spatial coverage of each dataset category. Black bars represent the availability of wave boundary and wind forcing data used in the simulations."*

Section 4.2 and 4.3: the authors discuss much about the seasonal variability of wind patterns and related water motions. I wonder how much thermal aspects may play a role in the reliability of spatial patterns of the latter, as stratification impacts the development of surface currents which in turn interacts with waves. The authors use a 2D model which can be reliable for the case of fully stratified conditions during winter, but in summer freshwater inputs from ice melt and higher solar radiation strengthen density stratification such that the 3D processes can become important and the simulated 2D field might not be as representative. Deception Island is semi-enclosed with steep bathymetry — a perfect environment for internal tide generation, which are driven by stratification. I'm not saying that the authors should run a 3D model, but I must say that such a long-term dataset without information on thermal dynamics reduces the range of potential studies using it, and the authors should maybe consider commenting these aspects somewhere.

We fully agree that thermal stratification and associated 3D processes may significantly influence hydrodynamic behavior, particularly in semi-enclosed basins such as Deception Island. During the austral summer, increased freshwater input from glacial melt combined with enhanced solar radiation can lead to strong vertical density gradients, affecting the vertical structure of currents and their interaction with wind and wave forcing. In contrast, the austral winter is typically characterized by more homogeneous water columns and well-mixed conditions, where depth-averaged modeling approaches are more representative.

While the current dataset was generated using a 2D (depth-averaged) configuration of the Delft3D-FLOW model, this decision was driven by the absence of vertical observational profiles (e.g., temperature, salinity, or current shear) necessary for calibrating and validating a 3D baroclinic model. Implementing a 3D model without proper vertical forcing and validation data would have introduced a high degree of uncertainty, potentially reducing the reliability of the results.

Figure 11. This visualization is rather unreadable and not so informative. It seems that the scope of the figure is comparing results in the different locations, so why don't the authors simply provide some aggregated information, e.g. pdfs from the three observation points?

The intention of this figure is to provide a representative example of the type of high-resolution temporal data available across different observation points in the domain. Aggregated information such as seasonal mean and monthly maxima have already been discused in previous figures for the entire region. We agree that more aggregated representations, such as probability density functions (PDFs), would be useful for detailed analyses; however, these fall beyond the illustrative scope of this manuscript. For users interested in such analysis, the

complete time series data for all observation points are openly available in the Zenodo repository. A sentence has been added to the manuscript to clarify this point.

*"The Figure \ref{fig_temp} serves as an example of the temporal structure and variability available in the dataset; full high-resolution time series for each observation point are openly available in the Zenodo repository."*

Figure 14: the velocity seems rather low if we are under an extreme event with about 13 m/s blowing in the caldera, but I guess that's because it is depth-averaged. However, what kind of output from Delft3D is presented here? It is depth-averaged velocity or only wave-induced velocity?

The current velocity shown in Figure 14 corresponds to the depth-averaged velocity field obtained from the 2D hydrodynamic simulation performed with Delft3D-FLOW. Since the model was run in depth-averaged, the output represents the integrated horizontal flow across the water column and includes the combined effect of wind stress, pressure gradients, and wave-induced forcing. We have clarified this in the figure caption and within the text to avoid confusion regarding the nature of the velocity data.

*"Hydrodynamic and atmospheric conditions during an extreme event on January 15, 2012, selected as a case study where the significant wave height exceeded the 99th percentile. Panels show: (a) significant wave height ($H_s$) [m], (b) peak wave direction [radians], and (c) depth-averaged current velocity [m/s] from the 2D Delft3D-FLOW simulation. This analysis provides insight into the spatial variability of wave and wind conditions during extreme events affecting Deception Island."*